# Bioanalysis of Oligonucleotide by LC–MS: Effects of Ion Pairing Regents and Recent Advances in Ion-Pairing-Free Analytical Strategies

**DOI:** 10.3390/ijms232415474

**Published:** 2022-12-07

**Authors:** Aowen Liu, Ming Cheng, Yixuan Zhou, Pan Deng

**Affiliations:** 1Jiangsu Key Laboratory of Neuropsychiatric Diseases and College of Pharmaceutical Sciences, Soochow University, Suzhou 215123, China; 2BioMarin Pharmaceutical Inc., San Rafael, CA 94949, USA

**Keywords:** oligonucleotide, RNA drugs, ASO, siRNA, bioanalysis, ion-paring, LC–MS, HILIC

## Abstract

Oligonucleotides (OGNs) are relatively new modalities that offer unique opportunities to expand the therapeutic targets. Reliable and high-throughput bioanalytical methods are pivotal for preclinical and clinical investigations of therapeutic OGNs. Liquid chromatography–mass spectrometry (LC–MS) is now evolving into being the method of choice for the bioanalysis of OGNs. Ion paring reversed-phase liquid chromatography (IP-RPLC) has been widely used in sample preparation and LC–MS analysis of OGNs; however, there are technical issues associated with these methods. IP-free methods, such as hydrophilic interaction liquid chromatography (HILIC) and anion-exchange techniques, have emerged as promising approaches for the bioanalysis of OGNs. In this review, the state-of-the-art IP-RPLC–MS bioanalytical methods of OGNs and their metabolites published in the past 10 years (2012–2022) are critically reviewed. Recent advances in IP-reagent-free LC–MS bioanalysis methods are discussed. Finally, we describe future opportunities for developing new methods that can be used for the comprehensive bioanalysis of OGNs.

## 1. Introduction

Therapeutic oligonucleotides (OGNs) have emerged as therapeutics with the ability to precisely and efficiently modulate gene expression [1,2]. As a new class of modalities, these molecules are under active development, including antisense oligonucleotides (ASOs), small interfering RNAs (siRNAs), aptamers, and microRNAs (miRNAs). Among those, single-strand ASOs and double-stranded siRNAs are the most advanced OGN therapeutics [3,4,5]. Modifications of the phosphodiester backbone, the ribose sugar moiety, and the nucleobase itself have been extensively attempted in order to improve in vivo stability against nucleases, as well as therapeutic efficacy [6]. To date (August 2022), a total number of 16 therapeutic OGNs have been approved by the U.S. Food and Drug Administration (FDA) and European Medicines Agency (EMA) (Table 1), including 10 ASOs, 5 siRNAs, and 1 aptamer. Among those, 13 OGNs therapies have been approved since 2016. Currently, therapeutic OGNs are being widely investigated to treat or prevent diseases that could not be addressed previously, and there are particular needs for powerful bioanalytical tools to study the pharmacokinetic and metabolism profiles of these molecules.

There are two major approaches currently available for OGN bioanalysis: hybridization-based immunoassays and liquid chromatography (LC) methods [26,27,28,29]. Hybridization ELISA methods are capable of quantifying OGNs in the pM range [30,31], and the formats used in OGN bioanalysis include one-step hybridization, two-step hybridization, dual ligation hybridization, and sandwich hybridization [27]. Locked nucleic acid (LNA) probes for hybridization could provide enhanced specificity [32,33], and the Meso-Scale Discovery (MSD) electro-chemiluminescent platform further improves detection sensitivity compared with the standard fluorescent reader [32]. Thayer et. al. reported a hybridization ELISA method using LNA probes and the MSD platform for the analysis of a siRNA, and the lower limit of quantification was achieved at 1.0 pM for both serum and liver samples [32]. LC-based assays offer several advantages: they are easier to develop, can selectively quantify the intact OGNs and their metabolites within a wide dynamic range, and are useful for the assessment of biodistribution. Traditionally, the sensitivity of the LC method (typically > 1 ng/mL) was not comparable with ELISA [34]. For the 16 FDA/EMA-approved therapeutic OGNs, LC–MS was utilized in the bioanalysis of 9 OGNs (Table 1), especially those approved in the most recent three years, for which LC–MS is an indispensable technology for PK profile and metabolite identification [16,17,19,20,21,22,23]. The reported values of the lower limit of detection (LLOQ) for LC–MS methods were at 10–20 ng/mL for golodirsen [16], givosiran [19,20], and vutrisiran [23], while the LLOQ was achieved at approximately 1 ng/mL by using an enzyme-linked immunosorbent assay (ELISA) (1.52 ng/mL for mipomersen [8,9,10], and 1 ng/mL for nusinersen [11], inotersen [14], and volanesorsen [15]). Comparisons of ELISA and LC–MS methods in the bioanalysis of OGNs can be found in Table 2. With the developments of LC–MS and sample extraction technologies, the current detection sensitivity can be achieved at sub-ng/mL levels in biological samples [35,36]. In particular, hybridization LC–MS combined the advantages of both ELISA and LC–MS platforms [37,38], with sample preparation being fulfilled by hybridization and quantitation conducted by IP-RPLC-MS. Targeted ASOs were extracted from biological samples (serum, plasma, cerebrospinal fluid, and tissues) by hybridization with biotinylated sense-strand OGNs coupled to streptavidin magnetic beads, and the extracts were analyzed by LC–MS with the LLOQ at 0.5 ng/mL [35,36]. Hybridization LC–MS provides high sensitivity as well as maintaining high specificity/selectivity from LC–MS, especially for truncated metabolites that typically introduce cross-reactivity issues in ELISA. Therefore, LC–MS currently represents an important technology for both qualitative and quantitative bioanalysis of OGNs.

Owing to their unique physiochemical properties, the bioanalytical method development for OGNs faces numerous challenges. OGNs are considerably more polar than other biomolecules with similar molecular weight due to their phosphate backbone. This hydrophilic property makes extracting and chromatographically retaining these molecules on conventional reversed-phase column extremely difficult. The multiple charge states of the OGN analyte and formation of cation adduct can complicate the mass spectra, hence decreasing sensitivity. The lack of retention and poor peak shape are usually observed by RPLC due to the extreme polarity of the analyte. The use of the IP reagent is the gold standard for improving ionization and reducing adduct formation, whereas the optimal IP reagent and modifier system can vary depending on different OGNs [39], necessitating a comprehensive understanding of the effects of IP reagents to facilitate screening of optimized experimental conditions. Although IP buffers can improve chromatographic performance as well as reduce MS charge state and alkali metal adducts [39,40], these advantages come at the expense of potential MS instrument contamination and decreased sensitivity [41]. In addition, it is a common practice to have dedicated systems for IP-RPLC–MS analysis of OGNs because the contamination occurs that could result in the ionization suppression when switching to positive polarity. To overcome this challenge, other methods such as hydrophilic interaction liquid chromatography (HILIC) have been explored over the years, and it has been proven to be a promising alternative to IP-RPLC in the bioanalysis of OGNs [42,43,44,45,46]. In the present paper, the recent development and applications of ion-paring and ion-paring-free bioanalysis strategies are reviewed. 

## 2. LC–MS/MS Bioanalytical Methods for OGNs Published during 2012–2022

We chose Web of Science Core Collection and PubMed to conduct the online retrieval of the required literature. The specific methods were as follows: The Web of Science Core Collection database searching language was “oligonucleotide (subject) (oligonucleotide includes antisense oligonucleotide, siRNA, miRNA, and aptamer) and bioanalysis (subject), or oligonucleotide (subject) and pharmacokinetics (subject)”, and the publication date ranged from 2012 to 2022. The PubMed database retrieval language was “oligonucleotide (Title/Abstract) AND bioanalysis (Title/Abstract)”, or “oligonucleotide (Title/Abstract) AND pharmacokinetics (Title/Abstract)”, and the publication date was limited to the most recent 10 years. All the literature available in full text in the two databases was collected, and 75 publications were obtained after deduplication. We then filtered the literature on the basis of the title and abstract of each paper, and 35 articles with the LC–MS bioanalysis method development were identified. Detailed information of the reported bioanalytical method is listed in Table 3, including targeted analytes, biological matrix, sample preparation, mobile phase, LC column, LC analysis time, LC–MS instrument, and lower limit of quantification (LLOQ).

## 3. Choice of Ion-Pairing Reagents to Improve LC–MS/MS Sensitivity

Electrospray ionization (ESI) operated under negative mode has been widely used in the MS analysis of OGNs. Concerns with ionization of OGNs are cation adduction and multiply charged precursor ions, which can lead to loss of MS sensitivity [75,76,77]. OGNs are adducted because the phosphodiester backbone is negatively charged in the solvent at neutral pH, and cations, such as Na^+^ and K^+^, are added to the backbone during ionization. These adducts are undesirable as they complicate MS spectrum and reduce the sensitivity by spreading the ion species into multiple *m/z* values. Electrospray creates a charge-state envelope that distributes total ion signal of the full-length oligonucleotide over multiply charged precursor ions, which can render the *m/z* of targeted OGNs into the MS detectable mass range. However, similar to the cation adduction, widely distributed multiply charged precursor ions can also lead to significant loss of detection sensitivity in LC–MS analysis. Therefore, the MS signal for a single OGN analyte can be distributed to a wide range of precursors ions, including alkali adducted, multiple charged ions, as well as the combinations of these two types of ions. It was reported that the degree of cation adduction increases with the length of the OGNs and is more extensive with phosphorothioate (PS) OGNs than with phosphodiesters [78]. Thus, a significant and general key for sensitive LC–MS analysis of OGNs is to obtain efficient ionizing by eliminating the formation of cation adducts and reducing multiple charges. 

Alkylamine reagents have been widely used in the LC–MS analysis of OGNs because they assist with the desorption of OGN ions into the gas phase during ionization, favor charge state reduction, and decrease cation adduction [79]. Traditionally, approaches have focused on the applications of triethylammonium acetate (TEAA) in the RPLC analysis of OGNs. However, TEAA is not particularly volatile and therefore not compatible with ESI-MS. Apffel firstly reported the application of trimethylamine (TEA)-hexafluoroisopropanol (HFIP) IP buffer in the analysis of OGNs, which provided both efficient ESI and good HPLC separation [76]. Since then, fluorinated alcohols such as hexafluoroisopropanol (HFIP) and hexafluoro-methyl-2-propanol (HFMIP) have been widely explored to improve LC–MS performance of OGNs. In addition, different alkylamine reagents were attempted as a IP reagent in the bioanalysis of OGNs (Table 4). 

Anecdotally, a sub-ng/mL LLOQ would typically be achieved by using an ELISA-based approach [27,80]. LLOQs in the single ng/mL range generally represent the technological limits of LC–MS detection [34]. In recent reports, sensitive LC–MS bioanalytical methods have been achieved with LLOQs well below 1 ng/mL [36,37,38,47,67]. It has been shown that the choice of ion-pair reagent can affect the degree of ion suppression and that the optimal ion-pair reagent and modifier system can depend on the type and content of the OGNs [39,81]. Previous studies have demonstrated that alkylamine IP agents produced the highest oligonucleotide MS signal intensity when used at concentrations around 15 mM [81,82]. Established bioanalytical methods for OGNs employed alkylamine IP agents at around 15 mM, including DMCHA [49], TEA [32,35,52], DIEA [61,62], DBA [67], and TPA [71]. In the analysis of trabedersen (PS-ASO), it was found that the optimum sensitivity was achieved using mobile phases containing 0.1% TEA (around 7.14 mM) and 1% HFIP (around 95 mM) as the IP modifiers. Higher concentrations of TEA-HFIP improved peak shape but significantly reduced the sensitivity [26]. In the analysis of mipomersen (PS and 2′-MOE modified ASO), the method development started with a mobile phase containing of TEA and HFIP at 7.1 and 68.8 mM, respectively. Although good peak shape and high sensitivity of the analyte were obtained, severe peak tailing and carry-over were observed after repeated injection of the processed plasma samples. In order to enhance the IP effects, the levels of TEA and HFIP were increased to 28.0 and 135.8 mM, respectively, and good peak shape was obtained; however, ion suppression was observed after such modification. The method was finally established at the expense of reduced sensitivity by using TEA/HFIP at 28.0/135.8 mM in mobile phase [47]. Therefore, in order to obtain a sensitive LC–MS assay, the composition of the ion-pairing reagents (types and concentrations of alkylamine and fluorinated alcohol) must be studied and optimized case by case, as the choice is heavily influenced by the targeted OGN. Researchers have investigated the effects of different alkylamine reagents and fluorinated alcohols on the MS response of OGNs [41,60,79,81]. McGinnis et al. evaluated 13 different alkylamines with HFIP and found that 10 mM diisopropylethylamine (DIEA) with 50 mM HFIP was preferred for hydrophobic phosphorothioate OGNs, while 10 mM diisopropylamine (DIPA) with 25 mM HFIP was ideal for hydrophilic siRNAs [81]. Studzinska et al. investigated the effects of IP reagents on the bioanalysis of phosphorothioate OGNs using LC–MS [60], including N,N-dimethyl-butylamine (DMBA), hexylamine (HA), N,N-diethylamine (DEA), and dibutylamine (DBA), which were used in the mobile phase at 5 mM and were mixed with 150 mM HFIP individually. It was found that the highest sensitivity was obtained by using DMBA for all three tested OGNs. In addition, 2.5 mM DMBA provided higher sensitivity compared with 5 mM DMBA in mobile phase. Basiri et al. investigated the role of five different fluorinated alcohols, as counterions for the cationic IP reagents, on the ESI-MS response of OGNs [79]. They found that by using a N,N-dimethylcyclohexylamine (DMCHA)/hexafluoro-2-methylisopropanol (HFMIP) mobile phase, the MS signal intensity of a phosphorothioate OGN could be enhanced significantly compared with any combinations of HFIP and alkylamine IP reagents. They also found that factors that influenced the ESI ionization efficiency of OGNs using alkylamine IP reagents included their boiling point, proton affinity, partition coefficient, water solubility, and Henry’s law constants [39]. They developed an algorithm (see below) to predict LC–MS signal intensity on the basis of the properties of selected IP reagents and composition of the targeted OGNs (contents of nucleotide), which was applied in the bioanalytical LC–MS method development of a microRNA (miR-451) [67], and it was predicted that miR-451 would generate the highest MS signal intensity in the presence of dibutylamine (DBA). Further optimization of the mobile phase using 15 mM DBA and 25 mM HFMIP resulted in a twofold increase in the MS response of the analyte compared with 25 mM HFIP [67]. 

Predicted LC–MS Signal Intensity = −0.00656 × Molecular Weight − 5.43532 × Density + 0.02322 × Boiling Point + 1.61079 × pKa − 0.12832 × Proton Affinity − 0.14625 × Gas Phase Basicity + 0.23521 Partition Coefficient + 0.00005 × Water Solubility − 0.00012 × Vapor Pressure + 0.00340 × Henry’s Law Constant + 4.75149 × Content A + 7.00368 × Content T + 4.39043 × Content C + 0.55245 × Content G + 47.72180 [39]

It has been recognized by many researchers that IP-RPLC–MS method sensitivity might decrease over time, especially in longer batches, which is considered as a result of mobile phase degradation and/or ion source contamination [37,38,59,83]. Although the signal intensity could be returned to its former level when the freshly prepared aqueous mobile phase was used, sensitivity decreases soon afterward, suggesting that there may be a solubility or stability problem with the aqueous mobile phase; this phenomenon was termed as “mobile phase aging” [83]. To ensure method ruggedness in handling long analytical batches, a ternary LC system and a short divert window into the ion source were employed. Li et al. developed a ternary pump system that was set up by 100% water as mobile phase A, 100% acetonitrile as mobile phase B, and an isocratic mobile phase C with 150 mM DMCHA and 250 mM HFMIP in acetonitrile at 10% of the total flow rate. This system provided stable detection sensitivity within long runs [37]. The method was further improved by introducing a short transfer window into the ion source, and this LC–MS method was capable of processing long analytical batches without significant loss of sensitivity [38]. 

## 4. Ion-Paring Reversed-Phase HPLC Separation of OGN and Its Metabolites

OGNs are hydrophilic analytes with limited retention on reversed-phase chromatography. IP-RPLC has been thus far the chromatographic choice for the bioanalysis of OGNs because of sufficient retention and resolution of OGNs [84,85]. IP reagents that are used in the bioanalysis of OGNs are listed in Table 4. In most cases, the IP-RPLC separations using TEA/HFIP mobile phase have been performed on bridged ethylene hybrid (BEH) or C18 silica stationary phase columns (Figure 1). IP-RPLC combines two retention mechanisms: ionic retentions introduced by alkylamine reagents and hydrophobic retention of OGNs on column sorbent (C18 in most cases) [85]. Donegan et al. investigated the effects of IP buffer on the separation of different categories of OGNs, including unmodified and modified OGNs such as 2′-F,2′-O-methylation (2′-OMe), N-acetylgalactosamine (GalNAc), and PS modification. They reported that alkylamine retention in RPLC correlates with the retention of OGNs in IP-RPLC, and hydrophobic alkylamine IP reagents provide better resolution of OGNs. In addition, they observed a positive correlation between retention on C18 column and the boiling points of alkylamines [85]. Therefore, boiling point could be a surrogate hydrophobicity that can be used for the selection of alkylamine reagents during method development. 

OGNs can be metabolized by nucleases generating truncated metabolites (n-1, n-2, etc.) [86]. These metabolites can potentially be differentiated from the full-length OGNs by chromatographic separation, MS, or both [64,84,87], which is the major advantage of the LC−MS method compared to ELISA. However, quantitation of metabolites using LC–MS can be challenging due to insufficient chromatographic separation and ion crosstalk between analytes [84]. For example, the 3′n- and 5′n-truncated metabolites could be isobaric pairs, which means that these metabolites are indistinguishable from each other by MS/MS alone. In other cases, the 3′n- and 5′n-truncated metabolites are not isobaric; however, a single alkali metal ion addition (Na^+^/K^+^) to a 3′n-1 precursor ion would result in a crossed ion with the 5′n-1 precursor ion. Therefore, chromatographic separation was expected to be essential [29,64,88]. Li et al. evaluated 15 alkylamines buffered with HFIP in the separation of full-length OGNs from their chain shortened n-1 analogs [40]. They proposed that the mechanism of retention with alkylamine at high concentrations (above 20 mM) was primarily micellar chromatography, while at lower concentrations of alkylamine, this changed to IP chromatography. Ewles et al. reported separation and quantification of an OGN and multiple metabolites using IP-RPLC–MS, including isobaric 3′- and 5′-truncated metabolites, which was achieved by using TEA/HFIP in mobile phase [26]. Li et al. reported an LC–MS method for the analysis of 20 mer ASOs and n-1 metabolite in plasma. Chromatographic separation of full-length ASO and metabolite was achieved by using a shallow gradient mobile phase with tributylamine (TBA) as the IP agent [37]. Basiri et al. reported that replacing TEA by dibutylamine (DBA) resulted in satisfactory retention and separation of miR-451 and its n-1 truncated metabolite [67]. 

Therapeutic OGNs are generally chemically modified to improve their biological stability and increase their in vivo half-life [6]. These modifications often make the molecule more hydrophobic and modestly reduce the bioanalytical challenge when compared to unmodified OGNs. GalNAc conjugation is a common modification to OGNs that could improve hepatic uptake of OGN therapeutics. Once distributed to the hepatocytes, GalNAc-OGN conjugate is designed to be rapidly cleaved to generate the deglycosylated (unconjugated) OGN, which is the pharmacologically active compound [89]. Ledvina et al. developed an LC–MS/MS method for the bioanalysis of a GalNAc-conjugated 16 mer OGN (AZD8233) using a mobile phase with IP buffer composed of ethylenediaminetetraacetic acid (EDTA)-TEA-HFIP. Unconjugated forms of AZD8233 (AZD8233-DG) were baseline separated from AZD8233 on a BEH C18 column, and therefore avoided crossed-ion interferences, and the LLOQ of AZD8233 was achieved at 0.200 ng/mL [36]. In a most recent report, by using capillary LC with HFIP and DIEA being employed as IP reagents, Husser et al. developed a highly sensitive IPRP-LC–MS method for the metabolite analysis of a GalNAc-conjugated ASO in hepatocyte, and the sensitivity was achieved at 0.8 ng/mL [73]. 

## 5. Ion-Pairing-Free LC–MS for the Bioanalysis of OGNs

IP reagents using in the RPLC are frequently associated with undesirable effects such as poor reproducibility, decreased column lifetimes, and increased instrumental downtime to clean ion-pairing residue in the LC–MS system. HILIC is an alternative method that can be used to analyze therapeutic OGNs. Recently, LC–MS analyses facilitated by HILIC in the absence of IP reagents has presented a promise for analyzing OGN therapeutics [42,90,91,92,93,94]. Studzinska et.al reported an IP-free investigation of various HILIC–MS conditions with a panel of PS OGNs extracted from serum, and LLOQ was achieved at 142–165 ng/mL [43]. MacNeill developed a HILIC–MS method for the analysis of an 18 mer OGN RM1 in human plasma, and the LLOQ was achieved at 10 nM [72]. The HILIC method was further applied to the bioanalysis of a 22 mer OGN (GNV705 AS) in cynomolgus plasma [70], and the LLOQ was established at 500 pM. Additionally, the n-1 and n-2 truncated metabolites were baseline resolved from the parent OGN under the HILIC conditions. 

Stationary phases of HILIC used for the bioanalysis of OGNs include diol (Phenomenex Luna HILIC) [43] and amide (Waters BEH amide [70,72], XBridge amide [72], TSKgel Amide [43], Amino-P-C2 [43]). Other types of HILIC columns, for example, those containing ionizable/zwitterionic groups [93,94] (ZIC-HILIC), have been used in the quality control analysis of OGNs; however, the application in the bioanalysis is yet to be explored. Demelenne et al. compared the performance of three kinds of HILIC columns in the separation of unmodified and PS-OGNs [93], and they found that in terms of the resolving power of targeted OGNs, amide and zwitterionic phosphorylcholine stationary phases outperformed the dihydroxypropane stationary phase. The mechanism of retention for different HILIC column chemistries is still under investigation. By using HILIC conditions, disadvantages associated with IP are avoided, such as prolonged instrumental downtime, decreased column lifetimes, difficulties in switching between positive and negative ionization modes, and system dedication to IP methods. 

Currently, the resolution of the HILIC approach is not comparable to the IP-RPLC method for the bioanalysis of OGNs. Kilanowska et al. compared the HILIC–MS with IP-RPLC–MS for the analysis of ASOs, and they found that IP-RPLC provided better results in terms of separation efficiency and MRM responses [44]. In the following metabolism study of ASOs in the human liver microsomes, they found that greater responses were obtained with the IP-RPLC method for most of the tested ASOs, including unmodified, phosphorothioate, and 2′-O-methyl modified ASOs [95]. However, these findings do not rule out the use of HILIC–MS as an alternative approach for OGN analysis, especially when a mobile phase free of IP reagents is required. 

## 6. Biological Sample Extraction

Sample preparation of OGNs is the most time-consuming and critical step for bioanalysis. Isolation of OGNs by using liquid–liquid extraction (LLE) combined with solid phase extraction (SPE) has been proven to be a useful approach [26,36,52,64,71]. In other applications, LLE alone with phenol/chloroform as the extraction solvent was also used for the bioanalysis of OGNs [43,60,61,65]. It has been well established that SPE alone could provide sufficient extraction recovery [47,49,54,70,72], which allows for potential high-throughput and automated robotic extraction. Most of the SPE extractions relied on reverse-phased sorbent (HLB [63,69], Strata X [51], and HySphere C18 [35]) and weak anion exchange (WAX) sorbent (Clarity OTX [47,49,54]), and the former is normally used in combination with IP regents. Among those SPE columns, Clarity OTX is most widely used in the bioanalysis of OGNs, as published in the last decade (Figure 1). In recent reviews, Nuckowski et al. summarized the different strategies applied for extraction of OGNs from biological matrices [96,97,98], indicating that Clarity OTX SPE tends to afford a higher recovery rate than HLB. To the best of our knowledge, the highest SPE recovery for therapeutic OGN was 91.8% for 2′-MOE-modified ASO mipomersen, as reported by Sun et al. [47], in which Clarity OTX SPE was used for the extraction of mipomersen from rat plasma. 

Other SPE method of OGNs have been explored in recent years [70,72,98]. MacNeill’s group developed a SPE method for the bioanalysis of 18 mer OGN in human plasma without IP reagents [72]. Waters Oasis WAX SPE column, a mixed-mode sorbent with cation exchange and reversed-phase moieties, was used. This study demonstrated that adequate acidification and sufficient sorbent capacity in the SPE load conditions are critical parameters for attaining best extraction recovery. To minimize the breakthrough of the OGN on the SPE column, 4.5% of phosphoric acid was used in plasma dilution, and a 10 mg sorbent column was chosen over the microelution format. The analyte was eluted with a mixture of acetonitrile and water (3:7, *v*/*v*) containing 2% ammonium hydroxide, and extraction recovery was achieved at 50%. In another study, this group developed a HILIC-SPE method using a NAX column (United Chemical Technology, Lewistown, PA, USA), which is an aminopropyl sorbent on a silica base [70]. The extraction recovery was achieved at 64.1%. No IP reagents were applied during the HILIC-SPE; therefore, the procedure is relatively quick, and simple reagents such as water, acetonitrile, ammonium hydroxide, and formic acid were used. 

Recently, the hybridization sample extraction method was combined with LC–MS detection to achieve sensitive and selective bioanalysis of therapeutic OGNs [37,38,48,50,53]. Li et al. developed a hybridization LC–MS method for the bioanalysis of a panel of ASOs [37]. Target ASOs were extracted from biological samples by hybridization with biotinylated sense-strand OGNs coupled to streptavidin magnetic beads, and LLOQ was achieved at 0.5 ng/mL using 100 μL of plasma. Extraction recovery was in the range of 89.9% and 109%. This method improved the sensitivity to a comparable level with ELISA. Dillen et al. developed a hybridization extraction approach to extract imetelstat from human and rat plasma by using a complementary biotinylated DNA probe, followed by detection using LC–MS [50]. The extraction recovery was 97.8% with the 5′-capture probe. Throughput is a major improvement of these methods, as the hybridization extractions can be processed in 96-well plate format and are highly automated. However, there are disadvantages to the hybridization extraction method. For examples, the extraction method relies on the hybridization affinity of the OGN analyte, as measured by melting temperature (Tm) against specific probes. As a result, the application of this method to metabolite analysis may be limited by low recovery, which may be improved by approaches to increase Tm. 

## 7. Conclusions and Future Perspective

This article summarizes the current applications of ion paring and ion-pairing-free techniques in the bioanalysis of OGNs. Advances in the IP-RPLC–MS method have led to sensitive methods with detection limits achieved at sub ng/mL, and selectivity has been enhanced with parent OGNs and truncated metabolites being simultaneously detected. However, there are still challenges in the development and applications of the IP-RPLC–MS method. The selection of IP buffers remains a trial-and-error process. HILIC–MS without IP reagents is a promising method, especially the amide column chemistry with the alkaline-buffered mobile phase and gradient conditions. However, this approach is currently held back by the relative low resolution, and further development of a HILIC sorbent suitable for OGN separation is needed.

Given the importance of distinguishing the full-length molecule from abundant nuclease-generated metabolites, the development of LC–MS assays for the simultaneous analysis of OGNs and their metabolites will accelerate in the upcoming years. Analysis of OGN metabolites is challenging because of the issues of crossed ions, most of which are due to the ions of analytes with similar *m/z*, for which unit-resolution triple–quadrupole MS is unable to differentiate. High-resolution quantitative MS is most likely to provide solutions to these problems, as it will distinguish analytes with similar *m/z* and thus avoid cross-ion interferences, as well as allow monitoring of all metabolites present, rather than only those specified by the multiple reaction monitoring method in triple quadrupole MS. Furthermore, retrospective analysis of the acquired high-resolution data could be used to identify unanticipated metabolites in incurred samples. 

There are now available extraction methods of for OGNs that are liberated from ion-pairing reagents, including SPE and hybridization methods. Much more work is needed to characterize the technique in terms of various SPE sorbent chemistries and extraction mechanisms. We anticipate that high-throughput IP-free analytical techniques will be further developed and adopted by the industry for the bioanalysis of OGNs in the coming years. 

## Figures and Tables

**Figure 1 ijms-23-15474-f001:**
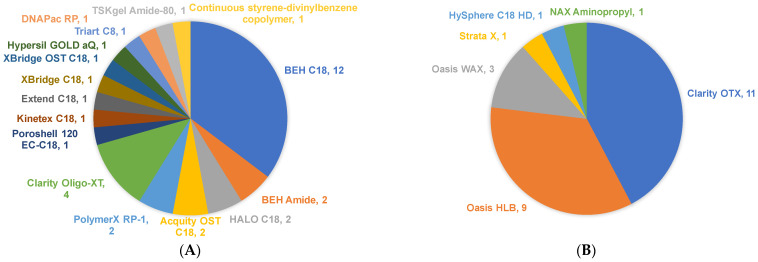
Different types of HPLC (**A**) and SPE (**B**) column sorbents used in the bioanalysis of OGNs. The number indicates the total applications of each sorbent. Data were summarized from a paper published from 2012 to August 2022. Detailed information is listed in Table 3.

**Table 1 ijms-23-15474-t001:** List of FDA- and EMA-approved oligonucleotide drugs and reported bioanalytical methods (until August 2022).

Category	Drug Name	Developer	FDA/EMA * Approval Year	Therapeutic Indication	Modification/Delivery	Length (nt)	Bioanalysis Method
ASO	Fomivirsen	Ionis Pharmaceuticals	1998	CMV retinitis in AIDS patients	PS	21	CGE-UV [7]
	Mipomersen	Ionis Pharmaceuticals	2013	Familial hypercholesterolemia	PS, 2′-MOE	20	CGE-UV [8,9], ELISA [8,9,10], LC–MS [9]
	Eteplirsen	Sarepta Therapeutics	2016	Duchenne muscular dystrophy	PMO	30	NA
	Nusinersen	Ionis Pharmaceuticals/Biogen	2016	Spinal muscular atrophy	PS, 2′-MOE	18	ECL, ELISA [11]
	Defibrotide	Jazz Pharmaceuticals	2016	Veno-occlusive disease in the liver		NA	LC–UV [12]
							LC–MS [13]
	Inotersen	Akcea Therapeutics and Ionis Pharmaceuticals	2018	Nerve damage in adults with hereditary transthyretin-mediated amyloidosis	PS, 2′-MOE	20	ELISA [14]
	Volanesorsen	Ionis Pharmaceuticals/Akcea	2019	Familial chylomicronemia syndrome	PS, 2′-MOE	20	ELISA, LC–MS, LSC [15]
	Golodirsen	Sarepta Therapeutics	2019	Duchenne muscular dystrophy	PMO	25	LC–MS [16]
	Viltolarsen	NS Pharma	2020	Duchenne muscular dystrophy	PMO	21	LC–MS [17]
	Casimersen	Sarepta Therapeutics	2021	Duchenne muscular dystrophy	PMO	22	NA
siRNA	Patisiran	Alnylam Pharmaceuticals	2018	Hereditary transthyretin-mediated amyloidosis	2′-OMe, LNP	21 (sense strand) + 21 (antisense strand)	LC-fluorescence [18]
	Givosiran	Alnylam Pharmaceuticals	2019	Acute hepatic porphyrias	PS, 2′-OMe, 2′-F, GalNAC	21 (sense strand) + 23 (antisense strand)	LC–HRMS [19,20]
	Inclisiran	Alnylam Pharmaceuticals/Novartis	2021	Hypercholesterolemia	2′-OMe, 2′-F, GalNAC	21 (sense strand) + 23 (antisense strand)	LC–HRMS [21]
	Lumasiran	Alnylam Pharmaceuticals	2020	Primary hyperoxaluria type 1	PS, 2′-OMe, 2′-F, GalNAC	21 (sense strand) + 23 (antisense strand)	LC–HRMS [22]
	Vutrisiran	Alnylam Pharmaceuticals	2022	Hereditary transthyretin-mediated amyloidosis	PS, 2′-OMe, 2′-F, GalNAC	21 (sense strand) + 23 (antisense strand)	LC–HRMS [23]
Aptamer	Pegaptanib	OSI Pharmaceuticals	2004	Neovascular age-related macular degeneration	PEG	28	LC-UV [24,25], ELISA [25]

*: Volanesorsen was approved by the EMA, and the others were approved by the U.S. FDA. 2′-F: 2′-fluorine; 2′-MOE: 2′-O-methoxyethyl; 2′-OMe: 2′-O-methylation; ASO: antisense oligonucleotide; CGE: capillary gel electrophoresis; CMV: cytomegalovirus; CSF: cerebrospinal fluid; ECL: electrochemiluminescence; ELISA: enzyme-linked immunosorbent assay; EMA: European Medicines Agency; FDA: Food and Drug Administration; GalNAC: N-acetylgalactosamine; LC–HRMS: liquid chromatography–high-resolution mass spectrometry; LC–MS: liquid chromatography–mass spectrometry; LSC: liquid scintillation counting; LNP: lipid nanoparticle; NA: not available; PEG: polyethylene glycol; PMO: phosphorodiamidate morpholino oligomer; PS: phosphorothioate; UV: ultraviolet.

**Table 2 ijms-23-15474-t002:** Strengths and limitations of ELISA and LC–MS for the bioanalysis of OGNs.

Assay	Strengths	Limitations
ELISA	High sensitivity (pg/mL LLOQ)	Narrow dynamic range
	No sample cleanup or extraction (except for tissues)	Needs specific capture/detection of probes, and assay development can be time consuming
	High throughput	Does not differentiate between intact and truncated species
LC–MS	High specificity	Less sensitive compared with ELISA (ng/mL or sub-ng/mL LLOQ)
	Wide dynamic range	Sample preparation can be time consuming and have less throughput
	Identification and quantification of truncated metabolites	Requirement of ion-pairing reagents for LC separation and retention

**Table 3 ijms-23-15474-t003:** Bioanalysis methods for therapeutic oligonucleotides published during 2012–2022 (until August 2022).

Analyte	Matrix	Sample Preparation	Mobile Phase	LC Analysis Time	LC Column	LC–MS Instrument	LLOQ
20 mer ASO [38]	Monkey serum	Magnetic bead extraction	A: H_2_O B: ACN C: ACN w/250 mM HFMIP/150 mM DMCHA (gradient elution)	7 min	Oligonucleotide BEH C18 (2.1 × 50 mm, 1.7 μm, Waters)	Shimadzu Nexera X2 UHPLC-Sciex 6500+ triple quadrupole	0.5 ng/mL
ASO [47]	Rat plasma	Clarity OTX SPE (Phenomenex)	A: H_2_O/TEA/HFIP (100:0.4:2) B: MeOH/TEA/HFIP (100:0.4:2) (gradient elution)	14.1 min	Triart metal-free C8 (2.1 × 100 mm, S-1.9 μm, 12 nm, YMC)	Thermo Fisher Scientific Vanquish UHPLC-Thermo Fisher Scientific Orbitrap HRMS Q Exactive Plus	0.5 ng/mL
ASOs and 3′n-1 metabolite [37,48]	Rat plasma and brain	Magnetic bead extraction	A: H_2_O B: ACN C: ACN w/250 mM HFMIP/150 mM DMCHA (gradient elution)	6.5 min	Clarity Oligo-XT (2.1 × 50 mm, 1.7 μm, Phenomenex)	Sciex ExionLC AD UHPLC-Sciex 6500+ triple quadrupole	0.5 ng/mL (plasma); 2.5 ng/g (brain tissue)
33 mer 2′-O-methyl modified PS-ASO [49]	Mouse and monkey tissues	Clarity OTX SPE (Phenomenex) or magnetic bead extraction	A: H_2_O/MeOH (95:5) w/15 mM DMCHA/25 mM HFIP B: H_2_O/MeOH (5:95) (gradient elution)	35 min	Clarity Oligo-XT (2.1 × 100 mm, 2.6 μm, Phenomenex)	Waters Acquity UPLC-Waters Synapt G2 Q-TOF	NA
13 mer OGN [50]	Rat and human plasma	Clarity OTX SPE (Phenomenex) or magnetic bead extraction	A: H_2_O/TEA/HFIP (100:0.2:0.2) B: MeOH/TEA/HFIP (100:0.2:0.2) C: THF/TEA/HFIP (100:0.2:0.2) (gradient elution)	5.01 min	Xbridge C18 (2.1 × 50 mm, 3.5 μm, Waters)	Shimadzu LC20AD HPLC-Sciex API4000 triple quadrupole	0.1 µg/mL (rat plasma) 0.5 µg/mL (human plasma)
ASO ISIS 681257 and metabolites [51]	Monkey plasma, urine, and tissues	LLE using phenol/chloroform/isoamyl alcohol (25:24:1) followed by Strata X SPE (Phenomenex)	NA	9 min	OST C18 (Waters)	Waters Acquity UPLC-Sciex 5500 triple quadrupole	1, 10, and 50 nM for plasma, urine, and tissues, respectively
ASO [52]	Mouse plasma and liver	LLE using phenol/chloroform (1:1) followed by Oasis HLB SPE (Waters)	A: H_2_O w/400 mM HFIP/15 mM TEA B: MPA/MeOH/ACN (2:1:1) (gradient elution)	8 min	Acquity BEH C18 (2.1 × 50 mm, 1.7 μm, Waters)	Sciex API5000 triple quadrupole	0.03 µg/mL (plasma); 0.03 µg/g (liver)
18 mer PS-OGN and 3′n-1 to n-3, 5′n-1 to n-3 metabolites [26]	Human plasma	LLE using phenol/chloroform/isoamyl alcohol (25:24:1) followed by Oasis HLB SPE (Waters)	A: H_2_O/HFIP/TEA (100:1:0.1) B: MeOH/HFIP/TEA (100:1:0.1) (gradient elution)	22 min	Acquity BEH C18 (2.1 × 100 mm, 1.7 μm, Waters)	Waters Acquity UHPLC-Sciex API5000 triple quadrupole	2 ng/mL
16 mer OGNs [53]	Rat and mouse plasma	Clarity OTX SPE or magnetic bead extraction	A: H_2_O/HFIP/TEA (100:0.5:0.2) B: ACN/isopropyl alcohol (95:5) (gradient elution)	10 min	DNAPac RP (2.1 × 50 mm, 4 μm, Thermo Scientific)	Shimadzu LC20AD HPLC-Sciex API4000 triple quadrupole	10 ng/mL
16 mer PS-ASO AZD8233 [36]	Human plasma	LLE using phenol/chloroform/ isoamyl alcohol (25:24:1) followed by Oasis HLB SPE (Waters)	A: (1mM EDTA/TEA (100:1))/TEA/H-FIP/H_2_O (2.5:0.25:1.25:100) B: (1mM EDTA/TEA (100:1))/TEA/H-FIP/MeOH (2.5:0.25:1.25: 100) (gradient elution)	5 min	Acquity UPLC BEH C18 (2.1 × 50 mm, 1.7 μm, Waters)	Shimadzu Nexera 30-series HPLC-Sciex 6500+ triple-quadrupole	0.2 ng/mL
10 mer GalNAc-OGN REVERSIR-A and metabolites [54]	Rat plasma and tissues	Clarity OTX SPE (Phenomenex)	A: H_2_O/HFIP/DIEA (100:1:0.1) w/10 μM EDTA B: ACN/H_2_O/HFIP/DIEA (65:35:0.75:0.0375) w/10 μM EDTA C: H_2_O/MeOH/ACN (10:45:45) (gradient elution)	29.8 min; 9.9 min	PolymerX RP-1 (2.0 × 50 mm, 5 μm, Phenomenex)	Thermo Fisher Scientific Dionex HPLC-Thermo Fisher Scientific Q Exactive	10 ng/mL (plasma), 100 ng/g (liver and kidney)
10 mer GalNAc-OGN REVERSIR-A and metabolites [55]	Monkey plasma, liver, and urine	Clarity OTX SPE (Phenomenex)	A: H_2_O/HFIP/DIE-A (100:1:0.1) w/10 μM EDTA B: H_2_O/ACN/HFIP/DIEA (35:65:0.75:0.0375) w/10 μM EDTA C: H_2_O/MeOH/ACN (10:45:45) (gradient elution)	29.8 min; 9.9 min	PolymerX RP-1 (2.0 × 50 mm, 5 μm, Phenomenex); Oligonucle-otide BEH C18 (2.1 × 50 mm, 1.7 μm, Waters)	Thermo Fisher Scientific Dionex UltiMate 3000 HPLC-Thermo Fisher Scientific Q Exactive	10 ng/mL (plasma and urine); 100 ng/g (liver)
22 mer GalNAc-siRNAs [56]	Rat plasma	Clarity OTX SPE (Phenomenex)	A: H_2_O/DIPA/HFI-P (100:0.15:0.264) B: H_2_O/MeOH/DIPA/HFIP (50:50:0.15:0.264) (gradient elution)	2.8 min	Acquity BEH C18 (2.1 × 50 mm, 1.7 μm, Waters)	Reversed-phase µHPLC-Sciex high resolution TripleTOF	10 ng/mL
siRNAs [32]	Rat and monkey tissues	Clarity OTX SPE (Phenomenex)	A: H_2_O w/15 mM TEA, 400 mM HFIP B: MeOH w/15 mM TEA, 400 mM HFIP (gradient elution)	14 min	Acquity UPLC Oligonucle-otide BEH C18 (2.1 × 50 mm, 1.7 μm, Waters)	Agilent 1290-Thermo Scientific Orbitrap Fusion Tribrid	NA
siRNA [57]	Rat plasma	Clarity OTX SPE (Phenomenex)	A: H_2_O w/15 mM TEA, 400 mM HFIP at pH 7.9 B: MPA/MeOH (50:50) (gradient elution)	8.1 min	Acquity BEH C18 (2.1 × 50 mm, 1.7 μm, Waters)	Waters Acquity UPLC-Sciex 5500; Waters XEVO TQ-S tandem quadruple and Waters Synapt G2-S Q-TOF	10 ng/mL
siRNA [58]	Human serum	NA	A: H_2_O w/400 mM HFIP, 16.3 mM TEA B:MeOH w/400 mM HFIP, 16.3 mM TEA (gradient elution)	34 min	Xbridge OST C18 (2.1 × 50 mm, 2.5 μm, Waters)	Shimadzu Prominence UFLC-Thermo Fisher Scientific LTQ Orbitrap	NA
PS-OGNs, PO-OGNs [59]	Mouse liver human and mouse liver microsomes	Clarity OTX SPE (Phenomenex)	A: H_2_O/MeOH (95:5) w/30 mM DMCHA/100 mM HFIP B: H_2_O/MeOH (5:95) (gradient elution)	23 min	Clarity Oligo-XT (2.1 × 100 mm, 2.6 μm, Phenomenex)	Waters Acquity UPLC-Waters Synapt G2 Q-TOF	NA
PS-OGNs OL1, OL8 [43]	Human serum	LLE using phenol/chloroform (1:1)	ACN/H_2_O containing 10–15 mM of AF (pH = 6.7) (56:44) (isocratic elution)	10 min	TSK gel Amide-80 (4.6 × 150 mm, 3 μm, Tosoh Bioscience)	Agilent 1100 HPLC-Agilent 6410 Triple Quad	142 ppb
PS-OGNs, metabolites [60]	Human serum	LLE using phenol/chloroform (1:1)	2.5 mM DMBA/150 mM HFIP and MeOH (gradient elution)	4 min	Hypersil GOLD aQ (2.1 × 100 mm, 1.9 μm, Thermo Scientific)	Dionex UltiMate 3000 UHPLC-Shimadzu LC–MS 8050	0.09 ng
24 mer PS-OGN and 3′n-1 [61]	Rat plasma	LLE using phenol/chloroform (2:1)	A: H_2_O w/15.7 mM DIEA/20 mM HFIP B: H_2_O/Et (50:50) w/15.7mM DIEA/20 mM HFIP (gradient elution)	5 min	Acquity BEH C18 (1.0 × 100 mm, 1.7 μm, Waters)	Waters Acquity UHPLC-Waters Synapt G2 Q-TOF	2.5 ng/mL
24 mer PS-OGN and 3′n-1 [62]	Rat plasma	Clarity OTX SPE (Phenomenex)	A: H_2_O w/15.7 mM DIEA/50 mM HFIP B: H_2_O/ACN (50:50) w/15.7 mM DIEA/50 mM HFIP (gradient elution)	9 min	Acquity BEH C18 (1.0 × 100 mm, 1.7 μm, Waters)	Waters Acquity UHPLC- Waters Synapt G2 Q-TOF	10 ng/mL
16 mer PS-OGNs [63]	Rat plasma	Oasis HLB SPE (Waters)	A: 10 mM CycHDMAA (pH 8.4), 100 µM ascorbic acid B: 10 mM CycHDMAA (pH 8.4), 100% ACN, 100 μM ascorbic acid (gradient elution)	15 min	Continuous styrene-divinylbenzene copolymer column (0.2 × 50 mm)	Fully integrated capillary HPLC-Sciex Q-TOF MS	100 nM
12 mer PS-OGN and 3′n-1 to n-3 and 5′n-1 [64]	Mouse plasma	LLE using phenol/trichloromethane (1:1) followed by Oasis HLB SPE (Waters)	ACN/0.05% aqueous NH_3_ (20:80) (isocratic elution)	2 min	Extend-C18 (2.1 × 150 mm, 3.5 μm, Agilent)	Agilent 1260 HPLC-Agilent 6410 Series Triple Quadrupole	20 ng/mL (parent), 10 ng/mL (metabolites)
2′-OMe, 2′-MOE, PS-modified OGNs, LNA, unmodified OGNs [65]	Human serum	LLE using phenol/chloroform/isoamyl alcohol (25:24:1)	5 mM DMBA,150 mM HFIP, MeOH (gradient elution)	10 min	Kinetex C18 (2.1 × 100 mm, 1.7 μm, Phenomenex)	Dionex UltiMate 3000 UHPLC-Shimadzu LC–MS 8050	0.15 µM
13 mer LNA miRNA [66]	Human plasma and urine	LLE using phenol/chloroform/isoamyl alcohol (25:24:1) followed by Oasis HLB SPE for plasma Oasis HLB SPE for urine (Waters)	A: H_2_O/HFIP/TEA (100:4:0.2) B: MeOH/HFIP/TEA (100:4:0.2) (gradient elution)	5 min	HALO C18 (2.1 × 50 mm, 2.7 μm, CPS Analitica)	Waters Acquity UPLC-Sciex API4000 (urine) and API5000 (plasma) triple-quadrupole	50 ng/mL
miRNA [67]	Rat plasma	Magnetic bead extraction	A: H_2_O/MeOH (95:5) w/15 mM DBA/25 mM HFMIP B: H_2_O/MeOH (5:95) (gradient elution)	12 min	Acquity UPLC BEH C18 (1.0 × 100 mm, 1.7 μm, Waters)	Waters Acquity UPLC-Waters Synapt G2 Q-TOF	0.5 ng/mL
13 mer miRNA [68]	Rat plasma	LLE using phenol/chloroform/isoamyl alcohol (25:24:1) followed by Oasis HLB SPE (Waters)	A: H_2_O/HFIP/TEA (100:4:0.2) B: MeOH/HFIP/TEA (100:4:0.2) (gradient elution)	3 min	HALO C18 (2.1 × 50 mm, 2.7 μm, CPS Analitica)	Waters Acquity UPLC-Sciex API5000 triple quadrupole	10 ng/mL
13 mer LNA miRNA [69]	Mouse plasma and monkey urine	µ-elution HLB SPE	NA	NA	NA	UHPLC-Sciex API4000 triple quadrupole	25 ng/mL (mouse plasma); 75 ng/mL (monkey urine)
22 mer RNA OGN and n-1 and n-2 metabolites [70]	Monkey plasma	NAX aminopropyl SPE (United Chemical Technology)	0.05% NH_4_OH in 10 mM AF and ACN (gradient elution)	6.5 min	BEH Amide (2.1 × 50 mm, 1.7 μm, Waters)	Waters Acquity UPLC-Sciex 6500+ triple-quadrupole, Sciex ZenoTOF 7600	500 pM
16 mer OGN [71]	Rat plasma	LLE using phenol/ dichloromethane (2:1); Oasis WAX SPE (Waters)	A: H_2_O w/15 mM TPA/50 mM HFIP B: MeOH w/20% MPA (gradient elution)	7 min	Oligonucleotide BEH C18 (2.1 × 100 mm, 1.7 μm, Waters)	Agilent 1290 UHPLC-Agilent 6490 triple quadrupole	0.25 nM
18 mer OGN [72]	Human plasma	Oasis WAX SPE (Waters)	0.02% NH_4_OH in 10 mM AF and ACN (gradient elution)	5 min	BEH Amide (2.1 × 50 mm, 1.7 μm, Waters)	Waters Acquity UPLC-Sciex 6500+ triple quadrupole	10 nM
15 mer unmodified OGN [35]	Human plasma	HySphere C18 HD online SPE (Spark Holland) Oasis WAX SPE (Waters)	A: H_2_O w/15 mM TEA/400 mM HFIP B: H_2_O/MeOH (50:50) w/15 mM TEA/400 mM HFIP(gradient elution)	7 min (online SPE-LC–MS/MS); 6 min (UHPLC)	EC-C18 Poroshell 120 (3 × 100 mm, 2.7 μm, Agilent) (online SPE LC–MS/MS); Acquity OST C18 (2.1 × 50 mm, 1.7 μm, Waters)	Waters Acquity UPLC-Sciex API5000 triple quadrupole	10 pM (online SPE-LC–MS/MS method) 0.05 nM (UHPLC–MS/MS method)
13 mer GalNAc-ASO [73]	Rat hepatocytes	Oasis HLB SPE (Waters)	A: H_2_O/MeOH/HFIP/DIEA (90:10:1:0.1) B: H_2_O/MeOH/HFIP/DIEA (10:90:1:0.1)	42.5 min	Xbridge BEH C18 (0.3 × 50 mm, 5 μm, Waters)	Thermo Scientific Dionex UltiMate NCP-3200RS HPLC-Thermo Scientific Orbitrap Fusion Tribrid	0.8 ng/mL
20 mer radavirsen [74]	Human plasma	Oasis µ-HLB SPE (Waters)	A: 1%FA in H_2_O B: ACN C: 1%FA in H_2_O, MeOH, and ACN (gradient elution)	3 min	Metasil AQ C18 (2.0 × 50 mm, Agilent)	Shimadzu HPLC system-AB Sciex API5000	5 ng/mL

2′-MOE: 2′-O-methoxyethyl; 2′-OMe: 2′-O-methylation; ACN: acetonitrile; AF: ammonium formate; ASO: antisense oligonucleotide; CSF: cerebral spinal fluid; CycHDMAA: cyclohexyldimethylammonium acetate; DIEA: N,N-diisopropylethylamine; DIPA: diisopropylamine; DMBA: N,N-dimethyl-butylamine; DMCHA: N,N-dimethylcyclohexylamine; EDTA: ethylene diamine tetraacetic acid; Et: ethanol; HFIP: hexafluoroisopropanol; HFMIP: hexafluoro-2-methylisopropanol; LLE: liquid–liquid extraction; LLOQ: lower limit of quantitation; LNA: locked nucleic acid; MeOH: methanol; miRNA: microRNA; MPA: mobile phase A; NA: not available; OGNs: oligonucleotides; PO: phosphodiester; PS: phosphorothioate; Q-TOF: quadrupole-time of flight; siRNA: small interfering RNA; SPE: solid-phase extraction; TEA: triethylamine; THF: tetrahydrofuran; TPA: tripropylamine; UHPLC: ultra-high-performance liquid chromatography.

**Table 4 ijms-23-15474-t004:** Alkylamines and fluorinated alcohols (organized in the order of increased boiling point) used in the ion pairing reversed-phase liquid chromatography–mass spectrometry.

Category	Name	Chemical Structure	Formula	Boiling Point (°C) *	LLOQ of Representative LC–MS Bioanalysis Method
Alkylamine	N,N-diethylamine (DEA)	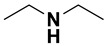	C_4_H_11_N	55.5	0.46–1.47 ng in serum [60]
Diisopropylamine (DIPA)	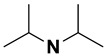	C_6_H_15_N	84	10 ng/mL in plasma [56]
Triethylamine (TEA)	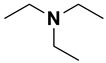	C_6_H_15_N	89.3	10 ng/mL in plasma [57]
N,N-Dimethyl-butylamine (DMBA)	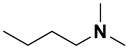	C_6_H_15_N	95	0.15 µM in serum [65]
N-Diisopropylethylamine (DIEA)	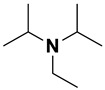	C_8_H_19_N	126.5	10 ng/mL in plasma [62]
Hexylamine (HA)	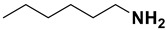	C_6_H_15_N	131–132	0.47–0.93 ng in serum [60]
Tripropylamine (TPA)	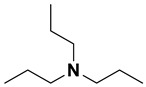	C_9_H_21_N	156	0.25 nM in plasma [71]
Dibutylamine (DBA)	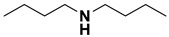	C_8_H_19_N	159–160	0.5 ng/mL in plasma [67]
N,N-Dimethylcyclohexylamine (DMCHA)	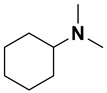	C_8_H_17_N	162–165	0.5 ng/mL in plasma [37]
Fluorinated alcohols	Hexafluoroisopropanol (HFIP)	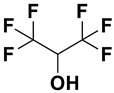	C_3_H_2_F_6_O	59	0.5 ng/mL in plasma [47]
Hexafluoro-2-methylisopropanol (HFMIP)	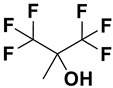	C_4_H_4_F_6_O	60.5–61.5	0.5 ng/mL in plasma [67]

* Boiling point data was obtained from SciFinder.

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
