# Peer review of "Bioanalysis of Oligonucleotide by LC–MS: Effects of Ion Pairing Regents and Recent Advances in Ion-Pairing-Free Analytical Strategies"

_ijms, 2022, doi:10.3390/ijms232415474_

Round 1

Reviewer 1 Report

A very well-written review by Liu et al highlights the recent developments and applications of ion-paring and ion-pairing free techniques in the bioanalysis of oligonucleotide based ASO/siRNA drugs. It provided a list of FDA-approved oligonucleotide-based drugs and their reported bioanalytical methods for pharmacokinetics profile and metabolite identification. Overall authors presented very detailed comparison from literature of IP-RPLC-MS method, and HILIC-MS IP free methods. It gives an idea to the readers about the real-world issues for the development of LC-MS assays for analysis of ASO/siRNA drug metabolites.

Minor correction:

Page 13 Fig 1.,

Please correct the annotation, it shows unwanted characters.

Author Response

Minor correction:

Page 13 Fig 1., Please correct the annotation, it shows unwanted characters

Response: We thank the reviewer for the very positive comments. Figure 1 has been corrected accordingly. 

Reviewer 2 Report

This manuscript provides an overview  of oligonucleotide bioanalysis by LC-MS, especially with regard to recent advances in ion pairing regents ion pairing-free analytical strategies. The topic is interesting and important, especially because the different modified oligonucleotides have emerged as therapeutics with the ability to precisely and efficiently modulate gene expression. Therefore, a study the pharmacokinetic and metabolism profiles of these molecules is particularly needed.

The authors showed sixteen therapeutic OGNs, which have been approved by US FDA (Table 1), and listed bioanalysis methods used for their analysis. In the introduction they try to shortly present two major approaches currently available for OGN bioanalysis: hybridization-based immunoassays and liquid chromatography (LC) methods. However this description is not very informative. A few sentences more should be added describing more precisely both methods (especially how ELISA can be used for analysis). For comparison it should be added also sensitivity of both methods (not only one of them! – line 5,p.3). Also more details should be added about hybridization LC-MS methods which was combination of both enzyme-linked immunosorbent assay (ELISA) and LC-MS platforms (line 8, p.3). To summarize, the introduction part should bring short but clear description of methods used for bioanalysis of OGNs with their advantages and disadvantages including ELISA.

In section 2, the authors clearly indicated the criterions of searching the  Web of Science Core Collection database and the results of bioanalysis methods used for therapeutic oligonucleotides published during 2012-2022 (until August 20220) were presented in Table2.

However formatting of Table1 and Table2 is inappropriate, for instance: table 1, column 2: the name “Pharmaceutical” and in the next line “s”. The whole word “Pharmaceuticals” should be in one line. In table 2 similar error, f.e.: “chlorofo” and “rm” in next line. The both tables should be also formatted more clearly, maybe lines between columns or rows.

The title of section 3 is “Choice of ion-pairing reagents to improve LC-MS/MS sensitivity”. However, no value is not presented on how this sensitivity can change/is improved using different IP. The terms ” higher concentrations” or “lower concentrations” are not informative. These descriptions are too general: what does higher means? Higher than what value? These values must be provided. This section presents a lot of generalities and little specificity. For example: It would be nice if an algorithm for selecting of IP reagent based on the composition of the targeted OGNs, which was applied in the bioanalytical LC-MS method development of a microRNA (miR-451) would be show.

The use of different alkylamine reagents as IPs  was shown in Table3, but references should be added for each IP, as well sensitivity (if possible) as done in table 2.

In concluding, the manuscript contain many remarkable information as well as interesting point of view but it should be filled by data indicated above.

Round 2

Reviewer 2 Report

The authors made proper changes according to the suggestions.

However I found some minor errors:

- in table 3 please be consistent and everywhere change "mice" to "mouse" (e.g.for 12-mer PS-OGN and 3´n-1 to n-3 and 5´n-1 [64]

- Figure 1 - there is something wrong in formatting in my copy of manuscript v2. In first version all was OK.
